# Examining outdoor play associations in Canadian early learning and child care centres: Cross-sectional insights from the Measuring Early Childhood Outside survey

Rachel Ramsden[1,2,3,4], Barry Forer[1,3], Hebah Hussaina[1,2], Christina Han[2,4], Caroline Bouchard[5], Jeff Crane[6‡], Megan McPhee[7‡], Michal Perlman[8‡], Mariana Brussoni[1,2,3,4*]

1 School of Population and Public Health, University of British Columbia, Vancouver, BC, Canada, 2 British Columbia Children's Hospital Research Institute, Vancouver, BC, Canada, 3 Human Early Learning Partnership, University of British Columbia, Vancouver, BC, Canada, 4 Department of Pediatrics, University of British Columbia, Vancouver, BC, Canada, 5 Faculté des sciences de l'éducation, Université Laval, Québec, Québec, Canada, 6 Faculty of Education, Memorial University of Newfoundland, St. John's, Newfoundland, Canada, 7 Early Childhood Development Association of PEI, Charlottetown, Prince Edward Island, Canada, 8 Department of Human Development and Applied Psychology, University of Toronto, Ontario, Canada

‡ These authors contributed equally to this work.
* mbrussoni@bcchr.ubc.ca

## Abstract

Canada lacks national data on the current provision of outdoor play (OP) in Early Learning and Child Care (ELCC) programs. In this study, we report results of the Measuring Early Childhood Outside (MECO) national survey to fill this gap and examine the factors that are associated with children's OP and risky play in ELCC programs. Respondents included ELCC centres providing full-day licensed group care (birth to school entry) in Canada. Primary outcomes measured were OP frequency, OP duration and risky play occurrence. Hierarchical multiple regressions were used to examine relationships and interaction effects between the primary outcomes and 14 variables encompassing centre, staff, physical environment and OP provision characteristics, for infant/toddler-aged and preschool-aged programs separately. A total of 1,187 ELCC centres responded to the MECO survey (9.8% response rate), of which 67.2% were non-profit providers. Most centres went outdoors every day, regardless of the season, though they spent less time outdoors in the winter than in the summer. Risky play was limited, with play at heights being the most common, and use of fire the least common. Variables that emerged as positively associated with most outcomes across programs related to training of centre directors and educators, giving children the autonomy to make decisions about going outdoors, providing all-weather gear, including diverse affordances (loose parts, gardening elements, fixed equipment), having outdoor spaces larger than required by licensing requirements, and the use of off-site spaces. Information about the current state of OP in ELCC

**Data availability statement:** All data are available in the UBC Research Data Collection at borealisdata.ca/dataverse/UBC_RD (doi: 10.5683/SP3/WSZ1W5).

**Funding:** This work was supported by the Lawson Foundation, the Lyle S. Hallman Foundation and the Muttart Foundation [grant number GRT 2022-10, 2022-2024]. We gratefully acknowledge their generous financial support. MB is supported by a salary award from the British Columbia Children's Hospital Research Institute.

**Competing interests:** MB is an unpaid board member of Outdoor Play Canada and occasionally undertakes consultancy work related to children's play. This does not alter our adherence to PLOS ONE policies on sharing data and materials.

centres is important at a time of considerable expansion in the sector, helping inform evidence-based policy development to enhance OP opportunities across Canada.

## Introduction

### Outdoor play and child development and well-being

Research outlines the profound benefits of outdoor play (OP) to multiple aspects of children's development, health and well-being [1–3]. OP differs from indoor play by offering its own distinct benefits, including being more inclined towards dynamic and vigorous physical activities [4,5]. Children are naturally driven by curiosity, and outdoor environments are the ideal settings for them to explore, engage their senses, and learn through hands-on experiences [3,6]. OP also nurtures essential skills that shape a child's development and sense of self [3], and invites open-ended interactions, spontaneity, risk-taking, exploration, discovery, and connection with nature. In addition, OP cultivates social and emotional growth by helping children build social skills, learn cooperation, and resolve conflicts effectively [7,8].

OP is also more likely to consist of risky play, a thrilling form of physical play involving uncertainty and the potential for injury [9,10]. Risky play includes: 1) Play with great heights; 2) Play with high speed; 3) Play with dangerous tools; 4) Play near dangerous elements; 5) Rough-and-tumble play; 6) Play where children go exploring alone; 7) Play with impact; and 8) Vicarious play [10,11]. Activities such as climbing, running, exploring independently, and rough-and-tumble play can contribute to physical activity and motor skill development, and cognitive development [9,12]. Risky play allows children to test their limits, assess risks, experience feelings of exhilaration and fear, and develop critical decision-making skills [8,12,13]. For example, deciding how high to climb or how fast to swing involves complex cognitive calculations related to capabilities, conditions and preferences [14]. Further, engaging in manageable levels of fear and stress helps children regulate emotions, such as fear and excitement, while building resilience and self-confidence [15]. By testing their limits and understanding their abilities in controlled settings, children gain a sense of mastery, modulate their emotions, and reduce fear responses in other situations – fostering essential life skills and emotional growth [13,15,16].

Three key ingredients – time, space and freedom – have been identified as necessary for OP opportunities [8]. *Time* involves making play a daily priority, as important as sleep or other basic functions. Prioritizing *time* for children's OP requires understanding the existing OP provision across families, structured settings (e.g., school, child care), and the wider community and seeking ways to combat the current displacement of time outdoors [17,18]. *Space* relates to ensuring that children have easy access to stimulating play spaces replete with diverse affordances for play. Supporting quality OP spaces for children includes considerations on play space design, such as the affordances provided and availability of loose parts, and inclusive and safe access [19–21]. *Freedom* involves allowing children to play in the ways they choose, including taking risks

that naturally emerge during play. Current evidence highlights the need to remove policy and structural barriers, and reframe risk to recognize the value of risky play, to create richer OP experiences that allow children greater freedom and opportunities for challenge [22–25].

Despite the numerous benefits, there is an evident decline in OP [26], especially in Western societies [8,21,27–29]. The time dedicated to OP has shrunk as structured academic activities indoors have been prioritized, and screens have become ever more enticing [26]. Outdoor spaces for play have become less accessible and appealing with increasing urbanization and traffic [30], and an emphasis on safety has led to play spaces with limited challenges [31,32]. Children's freedom and participation in risky play has been curtailed by risk-averse parents, educators and play providers concerned about injury and liability [27,33]. The noticeable decline in children's OP highlights critical opportunities to influence the *time, space* and *freedom* of OP provision in children's lives.

## Outdoor play in early learning and child care programs

Early learning and child care (ELCC) centres are important environments to examine with children under the age of 6 in many countries spending most of their waking hours in child care [34]. In Canada, 56% of children aged 0–5 years are in some form of child care, an increase from prior years [35]. The influence of time, space and freedom related to OP in ELCC programs can be mapped onto the various levels of Ecological Systems Theory [36,37]. At the societal (macrosystem) level, social and cultural norms have devalued the importance of OP within the ELCC sector, limiting *time* for play due to perceptions that time is better spent in academic preparation or indoor activities [38]. ELCC professionals may have limited experiences with the outdoors or negative attitudes toward OP, or face competing interests from scheduling and family influences, further reducing time allocated outdoors in ELCC centres [37,39]. Further, rapid urbanization at the societal level has put pressure on the *spaces* available for OP, with outdoor areas often the last to be prioritized in ELCC centres [40–42]. Thus, at the centre (microsystems) level, this means that many centres entirely lack outdoor spaces and in others, these spaces are often limited in size, poorly designed, and lack the natural elements that foster creativity, exploration, and diverse forms of play [43,44].

At the organizational and regulatory (exosystem) level, licensing and other policies have restricted *freedom* for OP [45]. Regulations and licensing requirements, while essential for safety, can have unintended consequences, particularly when they are interpreted in restrictive ways [37]. Playground safety standards can be excessively rigid and do not consider the local context and capabilities of the children playing in those spaces, leading to overly cautious designs that remove essential elements of challenge and exploration from children's play [23,37]. Additionally, the mesosystem level interaction between environments means that at the centre (microsystems) level, safety concerns and risk aversion among families and educators, compounded by educators' fears of liability, can make it difficult to balance safety with the need for children to experience challenge and risk-taking [37,46]. Furthermore, educators' perspectives and training play an important role, as some may not see the value of spending time outdoors or lack confidence in supervising OP or advocating for risk-taking activities, leading them to opt for "safer," indoor alternatives [37,45,47]. The balance between educator restriction, guidance, and scaffolding plays a pivotal role in shaping the degree of freedom afforded to children during OP experiences [48].

Despite the challenges, there are promising developments emerging within Canadian ELCC programs. Policymakers and practitioners are increasingly recognizing the need for a more balanced approach to facilitating risk and challenge in OP [45]. The importance of OP is gaining traction, with an emphasis on providing children with diverse play affordances, including risky play [49]. By offering consistent and extended access to OP throughout the day and across various seasons, ELCC programs have the unique potential to make OP a fundamental and routine component of children's daily lives [45]. Enhancing OP opportunities in ELCC programs plays a critical role in supporting children's health, well-being and development.

## Current state of outdoor play provision and policies in early learning and child care in Canada

The ELCC landscape in Canada is rapidly shifting and it is an opportune time to examine the current status of OP, as well as consider mechanisms for increasing access. The federal government of Canada launched two major initiatives: 1) the Multilateral Early Learning and Child Care Framework (MELCCF); and 2) the Canada-wide Early Learning and Child Care (CWELCC) plan [50,51]. Introduced in 2017, the MELCCF, guided by principles of quality, accessibility, affordability, flexibility, and inclusivity, aims to enhance ELCC services and their workforce, programs, and learning environments – particularly for children from Indigenous, rural and diverse needs backgrounds. Building on the MELCCF's foundation, the CWELCC was introduced in 2021 to provide funding to provinces and territories to support the expansion of child care services, enhance quality, and reduce fees for families, with the goal of lowering the average cost of child care to $10 per day by 2026. While these efforts represent significant strides towards addressing the child care needs of Canadian families, neither of these initiatives address children's OP provision and its relation to high-quality child care [42,52]. The rapid efforts to expand ELCC services can limit OP by exacerbating pressures on available outdoor spaces and compromising opportunities for educators to receive proper training on OP [53].

The current state of OP in ELCC in Canada is characterized by a complex and fragmented landscape, marked by regional inconsistencies in policies, regulations, and practices [54]. That said, provincial and territorial jurisdictions vary widely in their curricular emphasis on OP, with some program guides making minimal mention of it [55]. This disparity extends to the regulation of OP practices, with wide variations in required daily minutes, minimum outdoor space requirements, and safety standards [55]. For example, the minimum outdoor space requirement per child for infants differs widely between Alberta (2 $m^2$ per child) and Prince Edward Island (7 $m^2$ per child). Some jurisdictions do not require dedicated OP space connected to the ELCC centre, while others only require outdoor space for a percentage of participating children [55]. Further, licensing requirements vary regarding mandatory minimum spent outdoors, with some jurisdictions having no minimum requirement, and others, ranging from 60 minutes (British Columbia, New Brunswick) to 120 minutes daily (Ontario) [55]. Such inconsistencies contribute to a patchwork system where children's access to quality OP experiences is heavily dependent on their geographic location and the corresponding regulations.

The overall picture of OP in Canadian ELCC is one of ongoing evolution and adaptation. While significant challenges remain, an increased awareness of the importance of children's OP offers hope for a future where Canadian children have more equitable access to enriching and developmentally beneficial OP experiences. To support and enhance OP in Canadian ELCC settings, the existing political framework governing ELCC provision must prioritize OP opportunities within the federal, provincial and regional initiatives and regulations.

## Measuring Early Childhood Outside (MECO) pan-Canadian survey

The Measuring Early Childhood Outside (MECO) pan-Canadian survey was designed to capture the current provision of OP in ELCC programs in Canada, including the frequency and duration of OP, and permitted risky play activities. This national survey also strived to examine and measure theoretically relevant variables that are known to influence children's OP behaviour in ELCC environments. Existing evidence outlines centre-level predictors (i.e., auspice, geographic location and age groups served) [42], staff-related predictors (i.e., professional development, training and educator tenure) [37,39], physical environment-level predictors (i.e., OP play space size, loose parts, gardening areas and equipment types) [56], and provision-related predictors (i.e., all-weather gear and child autonomy) [37,47] of children's OP. Although these predictors shape opportunities for children's outdoor and risky play, their individual and combined influence on the current state of play provision in ELCC programs in Canada remains undiscovered. The MECO survey aimed to explore this relationship through the framework of time, space, and freedom (Fig 1).

Despite the growing recognition of the importance of OP in ELCC settings, we are aware of no Canadian data reporting on OP provision in these settings to-date. National data on the current status of OP in ELCC settings is imperative to

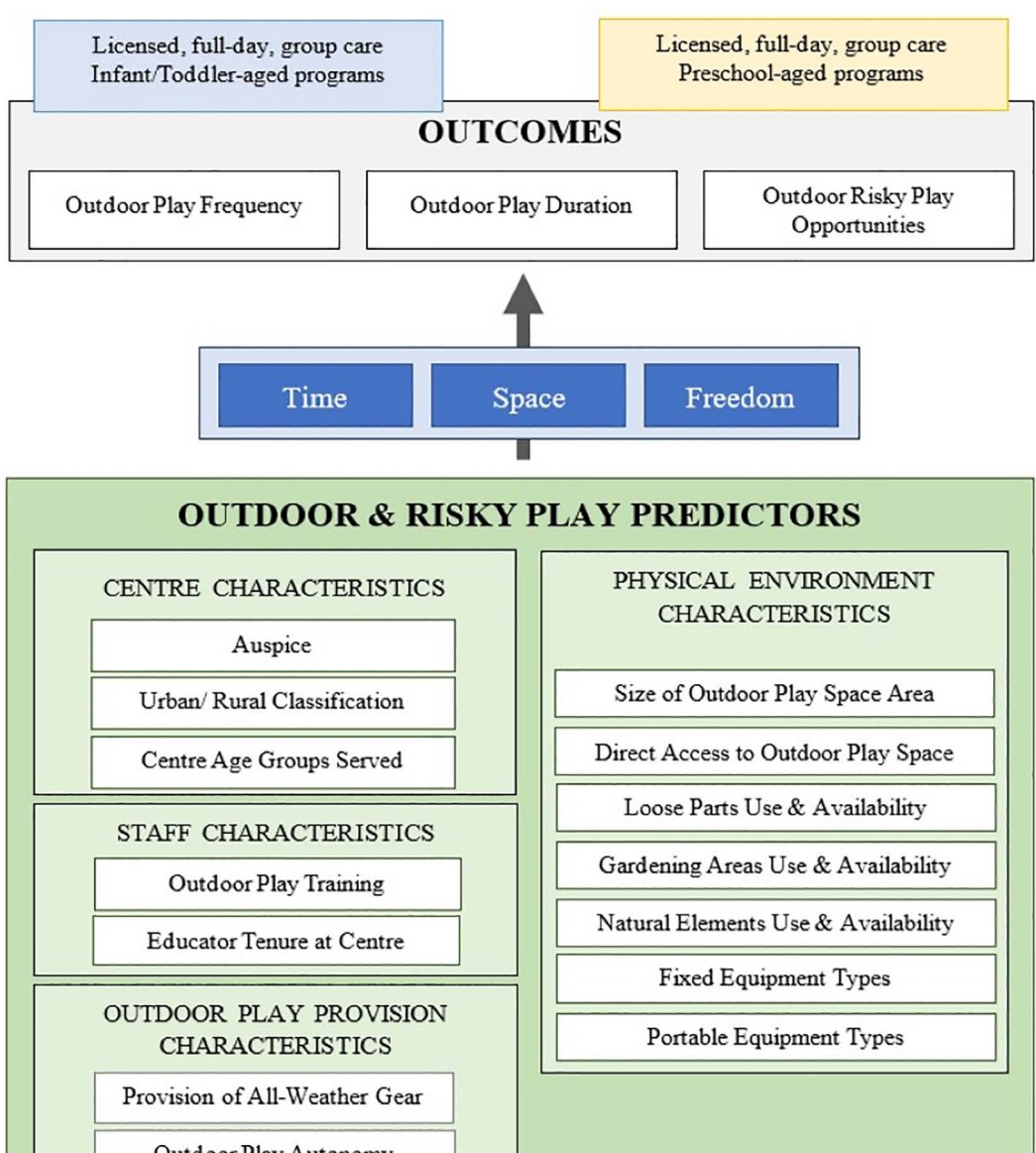

**Fig 1. A conceptual model of how centre, staff, physical environment and additional characteristics influence outdoor play and risky play outcomes through the constructs of time, space and freedom.**

identify gaps in the ELCC system, plan effective policies and strategies for improving OP opportunities, and contribute important knowledge to the research landscape. This paper aims to provide a cross-sectional overview on the current status of children's outdoor and risky play participation in Canada. Secondly, this paper assesses the association between important predictors of children's OP and children's outdoor and risky play outcomes. Investigating important predictors of children's OP supports an enhanced understanding of associations with children's OP, but does not make any causal claims on relationships.

## Materials and methods

### Setting and inclusion criteria

The target population for the MECO survey was ELCC centres in Canada providing full-day, group care to children before entry to primary school (birth to school entry age). This survey excluded nursery schools, school-age programs, pre-schools and other part-day programs, licensed family or in-home child care, and unlicensed programs due to the inconsistencies in outdoor programming and space-related regulatory standards within provincial and territorial policies. Limiting the survey to full-day, licensed ELCC group care centres allowed for a shared context for more detailed questions that were applicable to all respondents. Furthermore full-day group care ELCC centres deliver care to large groups of children and are expected to expand as an important child care option for Canadian families.

### Data collection procedure

**Survey development and pilot testing.** The MECO survey was developed with input from a literature review and an advisory committee. A literature review was conducted to find existing survey research that could inform survey development and survey questions previously used to measure children's OP provision. The study was guided by an advisory committee that included individuals from ELCC organizations and academic institutions across Canada. The committee set priorities for the survey, reviewed survey drafts, recommended recruitment strategies, and assisted with interpreting the results. The MECO survey was developed in English and subsequently translated into French. The English and French versions of the survey were hosted on REDCap [57], a secure, password-protected data collection platform hosted through the University of British Columbia. To ensure clarity and usability, the survey was pilot-tested with 9 ELCC providers (5 English-speaking and 4 French-speaking) through cognitive interviews. These interviews helped evaluate the content and comprehension of survey questions. Based on feedback from these sessions, the survey was iteratively revised. The final survey included eligibility screening questions and questions relating to the ELCC centre characteristics (e.g., auspice, staffing, size), program type, characteristics of the children attending the program, days and hours spent in OP (in summer and winter), factors influencing OP time (e.g., licensing regulations, weather, etc.), characteristics of the outdoor space (e.g., size, access, loose and fixed parts), permitted outdoor risky play activities, other outdoor locations, provision of all-weather gear, and staff training and experience. Respondents were also invited to upload photos of their outdoor space and share any other relevant information. The administered version of the MECO survey is provided in S1 Appendix.

**Survey distribution and promotion.** The University of British Columbia / Children's and Women's Health Centre of British Columbia Research Ethics Board provided ethics approval for the research (#H22-00210). The sampling strategy was designed as a convenience sample using publicly accessible ELCC program lists as the sampling frame. To distribute the survey, a comprehensive list of eligible ELCC programs across Canada was compiled from public databases, government contacts, and regional organizations. Where direct email contact information could not be obtained, outreach was done through trusted provincial or regional organizations. The survey was distributed via email to all provinces and territories, except Alberta and Ontario. In these two provinces, alternative strategies were used (e.g., social media promotion, collaboration with local advocacy groups, and posting in community newsletters) due to the lack of publicly available or easily reachable contacts. Additionally, no outreach was done to ELCCs in the Northwest Territories (NWT) as we were unable to obtain a research license to become eligible to collect data in the NWT. Social media promotion was initially employed but halted in August 2023 due to an influx of automated (i.e., 'bot') responses. Ongoing promotion included reminder emails sent to regional, provincial, territorial, and national organizations, and through newsletters such as the OP Canada Newsletter and e-blasts from the Canadian Child Care Federation. The survey was also promoted at the Breath of Fresh Air Conference in September 2023.

**Survey administration.** The MECO survey collected responses between June 23, 2023 and November 6, 2023 in both English and French. We requested that the survey be completed by only one respondent for each ELCC centre and required that the respondent was the individual most responsible for the day-to-day operations of the centre. If

respondents were responsible for the operation of multiple centres in separate physical addresses, they were instructed to complete a separate survey for each centre. Respondents who completed the survey were incentivized with a $25 gift card. To ensure data quality, enhanced security features were added to the survey (reCAPTCHA technology, email authentication, and 'honey pot' style verification questions tailored to ELCC-specific knowledge). Further, the research team validated responses by screening for automated bot submissions and duplicate responses. Examples of screening methods to eliminate automatic bot responses include examining overall survey response time (<3 minutes was determined as not possible for verified responses), reviewing responses to 'honey pot' questions, and examining open-text and photo submissions for artificially generated or stock responses. After the survey closed, verified responses were de-identified, assigned unique IDs, and exported from the REDCap platform for analysis.

## Survey response rates and weighting

A total of 1,187 respondents completed the MECO survey. Table 1 shows, for each province and the territories, the number of full-day group ELCC centres in the population, the number of responding centres, and the corresponding response rate. The total population was derived from the number of full-day child care centres published in the Early Childhood Education and Care in Canada 2021 report [58]. Overall, 9.8% of eligible ELCC centres responded to the survey. Four provinces had response rates above 20%, and five provinces had response rates below 10%. Due to low response rates, and ineligible participation from NWT, responses from Nunavut and Yukon territories were combined to make up the 'territories' jurisdiction. To rebalance these disproportionalities, responses were weighted while keeping the total number equal to the number of survey responses. As a result of weighting, the proportion of responses in each jurisdiction was restored to the population proportions, which sought to provide a more accurate estimate of results for Canada as a whole. While survey weighting sought to reduce bias from unequal regional representation, nevertheless a low overall survey response rate (9.8%) limits the generalizability of the survey findings to all Canadian ELCC centres.

## Analysis

**Outcome variables for outdoor play duration, outdoor play frequency, and risky play.** All the outcome variables described below were captured separately for infant/toddler (I/T) programs and preschool-aged (PS) programs – even if the centre served children in both age groups. Respondents self-identified the age categories (I/T and/or PS) their

**Table 1. MECO survey ELCC centre response rates, Canada and jurisdictions.**

| Jurisdiction | Population (Number of Centres in Canada) | Proportion of Total Canadian Population (%) | Sample (Number of Centres Responding) | Proportion of Total Survey Sample (%) | Survey Response Rate (%) |
|---|---|---|---|---|---|
| **Canada** | **12,076** | **100.0** | **1,187** | **100.0** | **9.8** |
| British Columbia | 2,117 | 17.5 | 466 | 39.3 | 22.0 |
| Alberta | 1,113 | 9.2 | 66 | 5.6 | 5.9 |
| Saskatchewan | 346 | 2.9 | 80 | 6.7 | 23.1 |
| Manitoba | 612 | 5.1 | 128 | 10.8 | 20.9 |
| Ontario | 3,312 | 27.4 | 154 | 13.0 | 4.6 |
| Quebec | 3,544 | 29.3 | 127 | 10.7 | 3.6 |
| New Brunswick | 422 | 3.5 | 40 | 0.3 | 9.5 |
| Nova Scotia | 271 | 2.2 | 100 | 8.4 | 36.9 |
| Prince Edward Island | 78 | 0.6 | 8 | 0.7 | 10.3 |
| Newfoundland and Labrador | 149 | 1.2 | 14 | 1.2 | 9.4 |
| Territories[a] | 112 | 0.9 | 4 | 0.3 | 3.5 |

[a]Population includes Nunavut, NWT and Yukon; Sample includes only Nunavut and Yukon.

program served, therefore the specific ages of each age category may differ by jurisdiction and as defined by regional licensing regulations. Seasonality was considered for the outcome variables OP duration (i.e., typical number of hours per day) and frequency (i.e., typical number of days per week), with responses captured separately for each season. For these analyses, responses from winter and summer were only considered to understand outcome differences between contrasting seasons. Altogether, eight outcomes were modeled: four for OP duration, two for frequency, and two for risky play (see below). Frequency of OP in the summer months was not modeled, as over 95% of centres reported daily frequency, leaving little remaining variability for modeling.

OP frequency ("In a typical week in each season, how many days are [the children] taken outdoors?" with response options from "0 days" to "5 or more days") was captured for each of the four combinations of season and age group of children. Responses were transformed into a quantitative value between 0 and 5, with the lowest response category ("0 days") assigned a score of 0 and the highest category ("5 or more days") assigned a score of 5. OP duration ("On a typical day in each season, approximately how many hours does each [child] spend outdoors"? with response options ranging from "none" to "5 or more hours") was captured for each of the four combinations of season and age group of children. Respondents answered based on a six-point scale, with a range of hours associated with each point. Duration responses were transformed to the mid-point of the range. For example, if a respondent chose "3 to 4 hours," their assigned numeric score was 3.5.

Risky play was assessed through four categories of activities – play at heights, play with tools, use fire, and rough-and-tumble play – in alignment with four of Sandseter's risky play categories [10,11]. Additional categories of risky play (play with high speed, play where children go exploring alone, play with impact, and vicarious play) were not measured in the MECO survey. For each of these activities, respondents were asked "How often do [children] at your program have opportunities to do the following activities outdoors?" For each item, respondents could select among five "how often" categories: 1) "not allowed", 2) "never", 3) "sometimes", 4) "often" and 5) "always." For the purpose of our analyses, "not allowed" and "never" were collapsed into one category. Assuming roughly equal intervals between the categories, an overall risky play score was calculated as the sum of the item scores, resulting in a total score ranging from 0 to 12. The Cronbach's alpha reliability for each derived risky play outcome scale (I/T and PS) was 0.67. While 0.70 is typically used as the threshold for acceptable scale reliability, for short scales and exploratory research, somewhat lower Cronbach alpha scores are considered acceptable [59]. This derived risky play score was used as the risky play outcome in our analyses. Unlike the OP duration and OP frequency outcomes, seasonality was not captured for the risky play outcome.

**Explanatory variables.** Fourteen variables were selected a priori for inclusion in multivariate regression modelling and grouped into four categories: 1) centre-level organizational characteristics; 2) staff-related characteristics; 3) physical environment characteristics; and 4) OP provision characteristics. An overview of all included explanatory variables are provided in Table 2 and further detail on how these variables were derived from the MECO survey is outlined in S2 Appendix.

## Analytical plan

Descriptive results are presented related to the number of respondents (N) for each variable. For each of the eight outcomes (summer OP duration, winter OP duration, winter OP frequency and risky play occurrence, for I/T and PS programs), a hierarchical multiple regression strategy was employed. Multiple models assessed the influence of each group of predictors on the outcomes, with the final model including all explanatory variables. A secondary model included interactions between staff training and each of three outdoor characteristics – loose parts, fixed equipment, and child autonomy (S3 Appendix). Interaction terms were included based on prior knowledge of the influence staff training has on additional predictors of children's OP, including loose parts inclusion [60], the use of fixed equipment or affordances [61,62], and children's autonomy in determining when they go outdoors [37]. Missing data were deleted listwise in the models. This resulted in a loss of only 5% of cases for the I/T models, and 6–7% of cases for the PS

**Table 2. List of 14 explanatory variables included in the analysis, summarized by category and variable levels.**

| Explanatory Variable | Category | Variable Levels |
|---|---|---|
| Auspice | Centre-level | 2 |
| Urban/Rural Classification | Centre-level | 2 |
| Centre Age Groups | Centre-level | 2 |
| OP Training | Staff-related | 4 |
| Educator Tenure at Centre | Staff-related | 3 |
| Size of OP Area | Physical Environment | 4 |
| Access to OP Area | Physical Environment | 2 |
| Count of Loose Parts | Physical Environment | 9 |
| Count of Gardening Areas | Physical Environment | 5 |
| Count of Natural Elements | Physical Environment | 7 |
| Count of Fixed Equipment Types | Physical Environment | 9 |
| Count of Portable Equipment Types | Physical Environment | 11 |
| Provision of All-Weather Gear | OP Provision | 3 |
| Children's OP Autonomy | OP Provision | 3 |

models. Given the large number of predictors (14 in each model), we assumed that the results would be minimally biased, even with listwise deletion. As suggested by Sawilowsky [63], betas of.200 or greater in the regression models were considered to reach the level of at least a small effect size. For all models and predictors, the Variable Inflation Factor was below 4.0, suggesting that multicollinearity was not a concern. The full results of these eight hierarchical regression models are located in S3 Appendix.

## Results

### Descriptive results for outcome variables

Weighted descriptive results for the three outcome variables modeled in this analysis are provided in Table 3. The results for these outcomes are separated for I/T and PS programs, and by summer and winter seasons for OP frequency and duration outcomes. Approximately 75% of responding centres offered both I/T- and PS-aged programs, therefore results separated by program type (I/T, PS) contain sample sizes larger than the 1,187 centres in the sample. For both I/T and PS programs, the mean OP frequency in summer is approximately five days per week (i.e., every day), therefore these outcomes were not modelled. I/T programs reported a mean OP frequency of 4.5 days per week outdoors in the winter, whereas PS programs reported a mean OP frequency of 2.9 days per week. To further understand the low reported OP frequency by PS programs, Table 4 provides OP frequency descriptive results by whether the centre serves only PS children or both age groups. The results show that the lower winter frequency of OP for PS children only applies when centres offer programs for both age groups. Both I/T and PS programs reported longer OP duration in the summer than in the winter months. The mean risky play score was 2.5 for I/T programs, and 3.8 for PS programs. Of the four activities, the one contributing the most to this difference between the age groups was the frequency of play at heights.

### Descriptive results for explanatory variables

Weighted descriptive results of investigated explanatory variables are summarized in Tables 5–7. Table 5 outlines the sample characteristics of respondents related to centre-level and staff-related characteristics. The weighted sample primarily consists of non-profit centres (67.2%), most often offering services to both I/T and PS children (75.4%), and located in urban locations (82.2%). Responding centres most frequently had no director or educators with OP-specific training

 

**Table 3. Weighted descriptive results for all outcome variables, by program type (infant/toddler & preschool-aged).**

| Program Type | Outcome Variable | Median | Mean | SD | Range | N |
|---|---|---|---|---|---|---|
| Infant/ Toddler | OP Frequency, Summer (days/ week) | 5.0 | 4.93 | 0.40 | 0–5 | 964 |
| | OP Frequency, Winter (days/ week) | 5.0 | 4.51 | 0.93 | 0–5 | 967 |
| | OP Duration, Summer (hours/ day) | 2.5 | 2.87 | 1.19 | 0–5 | 967 |
| | OP Duration, Winter (hours/ day) | 1.5 | 1.64 | 1.01 | 0–5 | 969 |
| | Play at Heights | 1.0 | 1.03 | 1.08 | 0–3 | 962 |
| | Play with Tools | 0.0 | 0.34 | 0.69 | 0–3 | 961 |
| | Use Fire | 0.0 | 0.07 | 0.35 | 0–3 | 963 |
| | Rough-and-tumble Play | 1.0 | 1.10 | 1.02 | 0–3 | 964 |
| | Risky Play (total score) | 2.0 | 2.52 | 2.24 | 0–12 | 967 |
| Preschool-aged | OP Frequency, Summer (days/ week) | 5.0 | 4.90 | 0.53 | 0–5 | 1,080 |
| | OP Frequency, Winter (days/ week) | 3.0 | 2.93 | 1.23 | 0–5 | 1,097 |
| | OP Duration, Summer (hours/ day) | 3.5 | 3.28 | 1.22 | 0–5 | 1,092 |
| | OP Duration, Winter (hours/ day) | 1.5 | 2.04 | 1.03 | 0–5 | 1,099 |
| | Play at Heights | 2.0 | 1.64 | 1.10 | 0–3 | 1,100 |
| | Play with Tools | 0.0 | 0.58 | 0.79 | 0–3 | 1,101 |
| | Use Fire | 0.0 | 0.11 | 0.42 | 0–3 | 1,096 |
| | Rough-and-tumble Play | 1.0 | 1.47 | 1.02 | 0–3 | 1,102 |
| | Risky Play (total score) | 4.0 | 3.79 | 2.36 | 0–12 | 1,105 |

**Table 4. Weighted descriptive results for outdoor play frequency in winter for preschool-aged programs, by age groups served at responding ELCC centre (weighted).**

| Age Groups Served | Outcome Variable | Median | Mean | SD | Range | N |
|---|---|---|---|---|---|---|
| Centre Serves Preschool-aged Children Only | OP Frequency, Winter (days/ week) | 5.0 | 4.47 | 1.02 | 1–5 | 211 |
| Centre Serves Infant/ Toddler-aged and Preschool-aged Children | OP Frequency, Winter (days/ week) | 2.0 | 2.56 | 0.96 | 0–5 | 886 |

(41.8%), or both the director and educators had received OP training (30.8%). Educator tenure across the responding centres varied, with the largest percentage of respondents (39.8%) indicating the majority of educators (>50%) had been at their ELCC centre at least five years.

Summarized weighted descriptive results related to physical environment and OP provision characteristics are provided in Table 6 (I/T programs) and Table 7 (PS programs). Most respondents reported that children have direct access to the OP areas from the indoor areas (81.9% and 78.5%, respectively), and their OP spaces were larger than required by licensing regulations (70.1% and 68.4%, respectively). The most-provided elements in OP areas for I/T and PS programs were portable equipment and loose parts. PS programs had higher mean scores than I/T programs for counts of loose parts, gardening areas, natural elements and fixed and portable equipment types. For both I/T and PS programs, almost half of respondents reported no provision of all-weather gear to either children or educators (46.0% and 48.5%). Where all-weather gear was provided, it was most often provided to children but not educators. Children in PS programs (17.6%) were more likely than children in I/T programs (14.5%) to have the autonomy to decide when to go to the OP area; however, the majority of programs indicated children's autonomy was rare or non-existent (43.6% and 51.1%).

## Multiple regression results across outdoor play outcomes

**Infant/toddler programs.** To understand the influence of the examined explanatory variables on OP opportunities in I/T programs, a final hierarchical regression model was employed for each outcome variable (Table 7). Type of auspice

**Table 5. Weighted descriptive results for centre-level and staff-related characteristics of responding ELCC centres.**

| Explanatory Variable | Total |
|---|---|
| **Centre Characteristic** | |
| Auspice [n(%)] | |
| For-Profit | 358 (30.1%) |
| Non-Profit | 798 (67.2%) |
| Indigenous-led Non-Profit[a] | 25 (2.1%) |
| Unknown[b] | 7 (0.5%) |
| Urban/Rural Classification [n(%)] | |
| Urban | 976 (82.2%) |
| Rural | 211 (17.8%) |
| Centre Age Groups [n(%)] | |
| I/T Only | 78 (6.6%) |
| PS Only | 214 (18.0%) |
| Both I/T and PS | 895 (75.4%) |
| **Staff-related Characteristic** | |
| OP Training [n(%)] | |
| None | 493 (41.8%) |
| Director Only | 211 (17.6%) |
| Educators Only | 111 (9.4%) |
| Educators and Director(s) | 363 (30.8%) |
| Proportion of Staff Educators with Five Years or Longer at the Centre [n(%)] | |
| Under 25% | 388 (33.4%) |
| 25% to 49.9% | 312 (26.9%) |
| 50% and Over | 462 (39.8%) |

[a]Included in non-profit category for all regression models.

[b]Not modelled.

had a small but significant effect on OP duration, with for-profit centres reporting longer OP duration in comparison to non-profit centres. Winter OP duration for I/T programs was shorter for centres serving both I/T and PS age groups ($\beta = -.075$) in comparison to centres serving only I/T aged children. Formal training in OP was associated with increased OP frequency in winter as well as more risky play, but only for centres where both the director and educators had OP training. The composition of educator tenure was only significantly associated with risky play. Although a small effect, higher proportions of long-tenured educators was negatively associated with risky play ($\beta = -.096$). Children's autonomy to go outdoors, both medium ($\beta = .165$) and high ($\beta = .230$), autonomy, were positively associated with increased risky play, as well as OP duration in both summer ($\beta = .066$) and winter ($\beta = .090$). The provision of all-weather gear to both children and educators was significantly related to greater duration of OP in both summer ($\beta = .076$) and winter ($\beta = .107$), and to more risky play ($\beta = .086$). Frequency of OP in winter was unrelated to gear provision for educators or children.

Characteristics relating to the physical environment of OP areas were associated with most examined I/T-aged program outcomes. Direct access to the OP area from the indoor program space was positively associated with summer OP duration ($\beta = .071$) and risky play ($\beta = .056$). Centres with larger OP areas than required by licensing regulations were significantly associated with increased OP frequency in the winter months ($\beta = .155$), and much larger OP areas were positively associated with summer OP duration ($\beta = .114$). I/T programs that relied on off-site OP areas to meet licensing requirements or had less than required on-site outdoor space were not significantly different than those with the minimum

**Table 6. Weighted descriptive results for physical environment and OP provision characteristics of responding ELCC centres, by program type (infant/toddler & preschool-aged).**

| Explanatory Variable | Infant/ Toddler -aged Total | Preschool-aged Total |
|---|---|---|
| **Physical Environment Characteristic** | | |
| Size of OP Area [n(%)] | | |
| Less Than Required or Off-Site | 143 (14.7%) | 212 (19.4%) |
| Exactly as Required | 147 (15.1%) | 133 (12.2%) |
| Slightly Larger Than Required | 331 (34.1%) | 318 (29.1%) |
| Much Larger Than Required | 350 (36.0%) | 429 (39.3%) |
| Direct Access to Outdoor Area [n(%)] | | |
| No | 174 (18.1%) | 236 (21.5%) |
| Yes | 788 (81.9%) | 861 (78.5%) |
| Outdoor Affordances [mean(SD)] | | |
| Loose Parts | 4.52 (1.69) | 4.53 (1.69) |
| Gardening Areas | 1.23 (1.16) | 1.45 (1.18) |
| Natural Elements | 2.67 (1.86) | 3.19 (1.83) |
| Fixed Equipment Types | 2.87 (1.54) | 3.44 (1.65) |
| Portable Equipment Types | 6.98 (1.83) | 7.28 (1.99) |
| **Outdoor Play Provision Characteristic** | | |
| Provision of All-Weather Gear [n(%)] | | |
| None | 446 (46.0%) | 535 (48.5%) |
| Educators Only | 7 (0.7%) | 10 (0.9%) |
| Children Only | 386 (39.8%) | 422 (38.3%) |
| Children and Educators | 131 (13.5%) | 135 (12.3%) |
| Children's OP Autonomy [n(%)] | | |
| Often/Always (High) | 141 (14.5%) | 194 (17.6%) |
| Sometimes/Occasionally (Medium) | 334 (34.4%) | 427 (38.8%) |
| Never/Rarely (Low) | 497 (51.1%) | 480 (43.6%) |

licensed outdoor areas for any of the four I/T outcomes. The count of loose parts was associated with increased OP frequency ($\beta = .109$) and OP duration ($\beta = .153$) in the winter. Additionally, OP duration in the summer ($\beta = .097$) and in the winter ($\beta = .118$) were positively associated with the number of gardening elements. The number of fixed equipment affordances within the OP area was significantly associated with I/T summer OP duration ($\beta = .153$). Risky play was the only outcome without an association with the number of loose parts, though there was an association with both the number of types of gardening elements ($\beta = .119$) and fixed equipment ($\beta = .138$).

Children went outdoors in the winter more frequently when educators and the director had OP training, regardless of the number of loose parts present. However, in ELCC centres where educators and directors had less OP training, increased access to loose parts was associated with more frequent OP during winter. This interaction highlights how OP training improves OP frequency in the winter months, and how access to loose parts improves this outcome. Centres where both educators and the director had OP training reported more loose parts on average (5.0 vs. 4.3). A hierarchical model outlining the results with interaction terms included is provided in S3 Appendix.

## Preschool-aged programs

To understand the influence of the examined explanatory variables on OP opportunities in PS programs, the same final hierarchical regression model used to investigate I/T outcomes was employed for each PS outcome variable (Table 8). Auspice was

**Table 7. Regression model results for ELCC centres serving infant/ toddler programs, all outcomes.**

| | | All Outcomes: Infant/Toddler Programs, Betas & Significance | | | |
|---|---|---|---|---|---|
| | | Duration, Summer | Duration, Winter | Frequency, Winter | Risky Play |
| Auspice | Non-profit vs. For-profit | **−.127**\*** | −.020 | .027 | −.056 |
| Rural/Urban | Rural vs. Urban Postal Code | .017 | −.038 | −.010 | .011 |
| Centre Ages | Both Ages vs. Just I/T | −.062 | **−.075*** | −.042 | −.042 |
| OP Training | Director Only vs. None | .05 | .044 | −.003 | .011 |
| | Educator Only vs. None | .05 | .027 | −.038 | .005 |
| | Director and Educators vs. None | −.008 | .043 | **.125**\*** | **.098**\* |
| Staff Tenure | Medium vs. Low | .02 | −.064 | −.013 | −.047 |
| | High vs. Low | .055 | −.064 | −.047 | **−.096**\* |
| OP Area Size | Less Than Required or Off-Site vs. Exactly as Required | .013 | −.033 | .028 | .061 |
| | Slightly Larger Than Required vs. Exactly as Required | .019 | .021 | **.155**\* | −.060 |
| | Much Larger Than Required vs. Exactly as Required | **.114*** | −.010 | .086 | .054 |
| Access to OP Area | Direct Access vs. Not Direct Access | **.071*** | −.045 | −.031 | **.056** |
| Affordances | Loose Parts | **.142**\*** | **.153**\*** | **.109*** | **.164**\*** |
| | Gardening Elements | **.097*** | **.118**\* | .021 | **.119**\* |
| | Natural Elements | −.041 | .046 | .062 | .008 |
| | Fixed Equipment | **.153**\*** | −.030 | −.045 | **.138**\*** |
| | Portable Equipment | −.039 | −.051 | .069 | −.030 |
| All-Weather Gear | Children or Educators vs. None | .058 | .036 | .039 | .043 |
| | Children and Educators vs. None | **.076*** | **.107**\* | .061 | **.086**\* |
| Child Autonomy | Medium vs. Low | **.128**\*** | .057 | .008 | **.165**\*** |
| | High vs. Low | **.066*** | **.090**\* | −.008 | **.230**\*** |
| | Adjusted R$^2$ | .135 | .076 | .067 | .254 |
| | N (weighted) | 920 | 922 | 920 | 921 |

\* $p < .05$, \*\* $p < .01$, \*\*\* $p < .001$.

statistically associated with summer OP duration, with non-profit centres reporting shorter durations of OP on average than for-profit centres ($\beta = −.121$). In addition, rural centres exhibited increased summer OP duration ($\beta = .063$) and risky play ($\beta = .058$). Responding centres with OP-specific training for both directors and educators had longer summer OP duration ($\beta = .081$) and increased risky play ($\beta = .092$). Educator OP training alone was also significantly associated with summer OP duration ($\beta = .080$) for PS programs. As was the case for I/T programs, there was a statistically significant negative effect for PS risky play and more educators with longer tenure ($\beta = −.075$). PS programs that provided children or educators with all-weather gear were positively associated with increased risky play ($\beta = .065$). Similar to I/T program results, child autonomy was strongly related to all four PS OP outcomes, with the largest effects for winter OP duration ($\beta = .153$) and risky play ($\beta = .242$).

PS programs with smaller than required OP spaces or that relied on off-site OP areas to meet licensing requirements had statistically longer OP duration in summer ($\beta = .168$), and more risky play ($\beta = .093$) than centres that met minimum on-site size requirements set by licensing (but did not exceed). Having much larger OP areas than required by licensing was also significantly associated with increased PS OP duration in the summer ($\beta = .218$). Of the five categories of OP affordances, the number of types of loose parts was most strongly related to the PS outcomes, reaching statistical significance for all four outcomes. The number of natural elements was positively associated with winter outcomes, both OP frequency ($\beta = .128$) and OP duration ($\beta = .095$). The number of types of fixed equipment was positively associated with the amount of risky play ($\beta = .146$).

**Table 8. Regression model results for ELCC centres serving preschool-aged programs, all outcomes.**

| | | All Outcomes: PS programs, Betas & Significance | | | |
|---|---|---|---|---|---|
| | | Duration Summer | Duration Winter | Frequency Winter | Risky Play |
| Auspice | Non-profit vs. For-profit | **−.121*** | −.058 | −.006 | .010 |
| Rural/Urban | Rural vs. Urban Postal Code | **.063*** | −.026 | −.031 | **.058*** |
| Centre Ages | Both Ages vs. Just PS | .029 | **.099*** | **−.612*** | **.082**** |
| OP Training | Director Only vs. None | .004 | .015 | .011 | .017 |
| | Educator Only vs. None | **.080*** | −.003 | −.021 | .045 |
| | Director and Educators vs. None | **.075*** | .023 | .032 | **.101**** |
| Staff Tenure | Medium vs. Low | .000 | −.021 | −.022 | **−.075*** |
| | High vs. Low | −.002 | −.018 | −.023 | −.035 |
| OP Area Size | Less Than Required or Off-Site vs. Exactly as Required | **.168*** | −.027 | −.010 | **.093*** |
| | Slightly Larger Than Required vs. Exactly as Required | .088 | .008 | −.027 | .018 |
| | Much Larger Than Required vs. Exactly as Required | **.218*** | .075 | .016 | .064 |
| Access to OP Area | Direct Access vs. Not Direct Access | .014 | −.054 | −.017 | −.002 |
| Affordances | Loose Parts | **.096*** | **.235*** | **.181*** | **.249*** |
| | Gardening Elements | .060 | .005 | −.033 | .067 |
| | Natural Elements | .043 | **.095*** | **.128*** | .074 |
| | Fixed Equipment | **.106*** | −.015 | −.030 | **.146*** |
| | Portable Equipment | .053 | −.051 | −.022 | −.065 |
| All-Weather Gear | Children or Educators vs. Neither | .039 | .056 | −.004 | **.065*** |
| | Children and Educators vs. Neither | −.002 | −.003 | −.009 | .0049 |
| Child Autonomy | Medium vs. Low | **.105*** | **.074*** | .043 | **.095*** |
| | High vs. Low | **.104*** | **.153*** | **.108*** | **.242*** |
| | Adjusted R² | .138 | .101 | .451 | .282 |
| | N (weighted) | 1,028 | 1,035 | 1,033 | 1,040 |

* $p < .05$, ** $p < .01$, *** $p < .001$.

A hierarchical model outlining the results with interaction terms included is provided in S3 Appendix. The largest interaction effect (β = .139) was between director and educator training and child autonomy, predicting winter OP duration. When both educators and directors had OP training, the mean winter OP duration was 2.2 hours per day, compared to 2.0 hours per day when only one or the other was trained.

## Discussion

The MECO Pan-Canadian survey fills an important data gap, delivering national data examining the existing provision of OP in Canadian ELCC settings. Findings from this study present a current snapshot of OP provision in Canadian ELCC programs, providing a baseline data point from which to assess future changes. These results also examine the factors that are associated with children's OP and risky play in ELCC programs to inform strategies and policies to enhance OP provision. The interpretation of these findings suggests some areas of strength within the Canadian ELCC landscape in relation to children's OP; I/T programs frequently reported daily OP in summer and winter seasons, and the majority of I/T and PS programs reported larger than required OP areas. However, the results from this study also highlight important areas for future attention and consideration, including limited opportunities for risky play and child autonomy within OP provision in Canadian ELCC programs. Below, we outline the current provision of OP within the context of *time, space* and *freedom*, and consider how these ingredients can be levers for change within the levels of the Ecological Systems Model as it relates to ELCC environments [36].

## Time

The MECO survey captured OP frequency and duration, revealing a distinct seasonal decline during winter across all programs. PS programs showed the greatest fluctuation, averaging 1.24 fewer hours and 2 fewer days outdoors per week in winter compared to summer. Comparatively, I/T programs reported, on average, similar decreases in OP duration between winter and summer (1.23 hours per day), but demonstrated minimal differences in seasonal OP frequency. Notably, mixed-age centres (comprising 75% of the sample) reported a median of only 2 days per week outdoors in winter among PS programs. The seasonal effect found in our data aligns with existing research highlighting reduced OP time in seasons that may experience more adverse weather conditions, such as snow, rain, wind and cold temperatures [64,65]. Seasonal trends in OP frequency and duration are also indicative of macrosystem level ideologies and approaches towards perceived adverse weather conditions that influence OP participation, as seen in previous studies [21,66–68]. While infant/toddler (I/T) programs spent more days outdoors in winter (4.5 days/week) than PS programs (2.9 days/week), this may reflect stroller walks rather than active outdoor play opportunities.

Educator training (microsystem and exosystem) and the provision of all-weather gear (exosystem) emerged as critical predictors of children's time outdoors in ELCC programs. The importance of training is well-recognized in helping ELCC professionals to build the skills to integrate OP into their practice [37,54]. ELCC centres where both the director and educators had OP training demonstrated more frequent OP for I/T programs in winter, and more time spent in OP for PS programs in summer. Furthermore, providing gear for both children and educators significantly extended winter OP duration, directly addressing common weather-related barriers [37]. These findings underscore the need for exosystem-level support, including government funding and institutional policies, to standardize professional development and the provision of all-weather clothing.

## Space

Characteristics of OP environments, including the size, accessibility and available affordances, such as loose parts and natural elements, have been previously identified as modifiable components of the ELCC microsystem related to children's OP provision [43,69,70]. Likewise, survey results indicated that the availability of loose parts was a critical variable of extended OP duration in both summer and winter. In addition, variety and availability of gardening elements and fixed equipment increased risky play opportunities, aligning with existing research on the importance of loose parts, natural elements and fixed equipment [60,71–73]. Interestingly, gardening elements specifically increased OP duration for I/T programs in both seasons, whereas natural elements (e.g., trees, shrubs, etc.) positively influenced PS programs in the winter. These findings may be due to I/T programs using gardening elements as a structured way to interact with the natural environment. While most programs reported high counts of portable equipment (e.g., tricycles, tables), these items did not enhance OP outcomes.

Overall, having an on-site area exceeding licensing requirements was associated with increased OP frequency and duration for both I/T and PS programs. Direct access to the OP area was associated with increased OP duration in the summer and risky play for I/T programs, however it was not related to any outcomes for PS programs. Furthermore, beyond the fence off-site excursions provided PS children with longer summer OP durations and greater risky play opportunities by offering diverse physical challenges and expansive natural environments. While, existing evidence outlines that OP areas adjacent to the indoor space are associated with enhanced OP opportunities in ELCC programs [74,75], our findings suggest that incorporating off-site areas can mitigate microsystem limitations. Off-site environments may offer different affordances then ELCC on-site outdoor areas, including more expansive spaces, other natural elements, and diverse physical challenges that promote exploration and creativity.

## Freedom

Children's freedom in outdoor play (OP) was assessed through risky play participation and autonomy. Risky play remains limited in Canadian ELCC centers, reflecting macrosystem-level concerns regarding injury and litigation [37,76,77]. Play

at heights and rough-and-tumble play were the most common, while use fire and play with tools were the least common types of risky play in I/T and PS programs. Overall, PS programs engaged in more risky play than I/T programs, particularly play at heights. Similarly, Sandseter et al.'s study of Norwegian ELCC centres reported age as positively associated with total amount of risky play and demonstrated that play at heights was among the most prevalent forms of outdoor risky play, however rough-and-tumble play was not as prominent in their sample [77]. Notably, child autonomy emerged as one of the strongest correlates for all OP outcomes, emphasizing the importance of allowing children, including infants, to decide when and how they play outdoors. Continuous access to outdoor environments and free routines have been previously shown to improve educator-child interactions [78] and increase physical activity, play and learning [78–80].

Educator training was associated with allowing PS children more autonomy in their OP choice. Programs where both directors and educators received training showed the highest levels of risky play, suggesting that a shared professional approach is more effective than isolated staff training [37]. In addition, a higher proportion of long-tenured educators was associated with lower levels of risky play, potentially reflecting entrenched mindsets and a lack of exposure to recent risky play pedagogy [9,54]. These findings highlight the need for accessible, ongoing professional development and post-secondary integration of OP training. Such initiatives help align microsystem practices with modern approaches and facilitate the communication of OP benefits to families and colleagues (mesosystem), including families and colleagues [66,81,82].

## Limitations

While the MECO survey offers valuable insights into Canadian ELCC OP provision, several limitations exist. The study focused exclusively on regulated, full-day centre-based ELCC settings, excluding nursery schools, licensed family child care, and programs providing only part-day or school-age care, which limits the generalizability of the findings across all care types. Future research could expand the sample to include part-day programs and family child care programs, providing a broader understanding of OP practices across different types of ELCC environments. Furthermore, the 9.8% response rate and regional inconsistencies, stemming from distribution challenges in Alberta, Ontario, and the NWT, impacted national representation. Notably, for-profit centers were underrepresented at 30% of the sample compared to the national 50% average [83]. While response weighting by province and territory sought to address the variance in response rates by geographic region, a higher overall sample size is required to assume generalizability of these results to the Canadian context. Future research should utilize robust engagement strategies, such as early partnerships with provincial regulatory bodies or regional academic institutions, to ensure more comprehensive coverage.

Data collection was further complicated by automated bot responses from social media promotion. Although rigorous validation measures were implemented to exclude fraudulent data, the necessary cessation of social media outreach likely restricted respondent reach in certain provinces. Future surveys should implement advanced verification techniques such as real-time automated response pattern monitoring. Finally, as a cross-sectional survey designed to establish baseline data, these findings illustrate patterns of association rather than causal inferences. This research serves as a critical foundation for tracking longitudinal progress and guiding more rigorous future inquiries into OP provision in Canada.

## Conclusions

The MECO survey provides insights into the current status of OP in ELCC centres at a timely moment in the history of child care in Canada. By situating the MECO survey within Canada's rapidly evolving ELCC landscape, this research highlights a critical opportunity to integrate OP into current federal and regional expansions. Despite significant gains in child care affordability and availability, OP remains under-prioritized; however, by addressing the interplay between institutional policy, educator training, and societal perceptions, stakeholders can develop comprehensive strategies to foster supportive environments. To this end, the study proposes actionable recommendations, including the provision of all-weather gear, enhanced pre-service and professional development on risky play, and the continuous improvement of outdoor space design through the use of natural features and loose parts. Ultimately, establishing the MECO survey as a

longitudinal benchmarking tool will provide the data-driven insights necessary to shift societal focus away from risk-aversion and toward a model that prioritizes the essential ingredients of time, space, and freedom for children's play.

## Supporting information

**S1 Appendix. MECO survey development, testing and administration.**
(PDF)

**S2 Appendix. Selection of explanatory variables.**
(DOCX)

**S3 Appendix. Hierarchical multiple regression models for all outcomes.**
(DOCX)

## Author contributions

**Conceptualization:** Rachel Ramsden, Christina Han, Mariana Brussoni.

**Data curation:** Rachel Ramsden, Hebah Hussaina, Mariana Brussoni.

**Formal analysis:** Barry Forer, Mariana Brussoni.

**Funding acquisition:** Christina Han, Mariana Brussoni.

**Investigation:** Barry Forer, Hebah Hussaina, Mariana Brussoni.

**Methodology:** Rachel Ramsden, Barry Forer, Hebah Hussaina, Christina Han, Caroline Bouchard, Mariana Brussoni.

**Project administration:** Rachel Ramsden, Christina Han, Mariana Brussoni.

**Resources:** Caroline Bouchard, Mariana Brussoni.

**Software:** Mariana Brussoni.

**Supervision:** Mariana Brussoni.

**Validation:** Rachel Ramsden, Barry Forer, Caroline Bouchard, Mariana Brussoni.

**Visualization:** Rachel Ramsden, Mariana Brussoni.

**Writing – original draft:** Rachel Ramsden, Barry Forer, Mariana Brussoni.

**Writing – review & editing:** Rachel Ramsden, Barry Forer, Hebah Hussaina, Christina Han, Caroline Bouchard, Jeff Crane, Megan McPhee, Michal Perlman, Mariana Brussoni.

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
