## [Decision Letter · Decision Letter 0]

23 Nov 2025

Dear Dr. Brussoni,

Thank you for submitting your manuscript to PLOS ONE. After careful consideration, we feel that it has merit but does not fully meet PLOS ONE’s publication criteria as it currently stands. Therefore, we invite you to submit a revised version of the manuscript that addresses the points raised during the review process.

We look forward to receiving your revised manuscript.

Kind regards,

Yih-Kuen Jan, PhD

Academic Editor

PLOS ONE

Journal Requirements:

Reviewers' comments:

Reviewer's Responses to Questions

**Comments to the Author**

1. Is the manuscript technically sound, and do the data support the conclusions?

Reviewer #1: Partly

Reviewer #2: Yes

2. Has the statistical analysis been performed appropriately and rigorously?

Reviewer #1: Yes

Reviewer #2: Yes

3. Have the authors made all data underlying the findings in their manuscript fully available?

Reviewer #1: Yes

Reviewer #2: Yes

4. Is the manuscript presented in an intelligible fashion and written in standard English?

Reviewer #1: Yes

Reviewer #2: Yes

Reviewer #1: This manuscript presents a well-designed and carefully written national survey that provides the first pan-Canadian data on outdoor play provision in Early Learning and Child Care centres. It addresses an important gap in the Canadian early childhood education and policy landscape. The conceptual framing around “time, space, and freedom,” grounded in ecological systems theory, is strong and appropriate. The manuscript is well organized and generally clear. However, several methodological limitations and areas requiring greater transparency should be addressed before the paper can be considered for publication.

The study design and rationale are appropriate, and the conclusions are largely supported by the data presented. The sampling frame, use of weighting, and regression analyses are suitable for an exploratory, cross-sectional study. Nonetheless, the low and uneven provincial response rate (9.8%), exclusion of certain jurisdictions, and reliance on weighting raise concerns about representativeness. These issues should be discussed more explicitly as potential sources of bias, and the findings should be interpreted as associative rather than generalizable to all Canadian ELCC settings.

The use of hierarchical multiple regression is appropriate for examining associations among the predictors, and the inclusion of interaction terms (training × affordances/autonomy) adds value. However, several analytic details need clarification:

1. Describe how missing data were handled and report the sample size used for each model after exclusions.

2. Provide checks for multicollinearity and model diagnostics (e.g., residual analysis, VIFs).

3. Justify treating ordinal risky-play variables as continuous and consider presenting sensitivity or robustness checks.

4. Including marginal-effect or predicted-value plots for key interactions would improve interpretability.

Tables are comprehensive and informative. Minor editorial improvements could enhance readability:

1. Ensure consistent terminology throughout (e.g., “off-site” vs. “larger-than-required” outdoor play areas).

2. Review for minor formatting or spacing inconsistencies.

3. Consider condensing portions of the Introduction that repeat background literature to improve flow and focus.

Reviewer #2: This manuscript explores the relationship between outdoor play (OP) and child development and well-being, providing a comprehensive and highly relevant analysis of the topic. The study systematically reviews the benefits of OP and examines its current status and challenges in early learning and child care (ELCC) environments in Canada. The research offers valuable insights for future studies and policy development. Below are the key strengths and areas for improvement identified in the manuscript.

1.Clarification of the "Time, Space, and Freedom" Framework:

While the "time, space, and freedom" framework is conceptually strong, its discussion in the manuscript is somewhat abstract. The authors are encouraged to provide concrete examples or case studies to illustrate how these factors influence the implementation of OP in real-world settings. Additionally, linking this framework more explicitly to the findings of the MECO survey would enhance its practical relevance.

2.Detailed Description of MECO Survey Methods:

The manuscript could benefit from a more detailed explanation of the MECO survey's data collection and analysis methods. Specifically, the authors should clarify the sampling strategy, the representativeness of the sample, and the measures taken to ensure the reliability and validity of the data. This would strengthen the transparency and credibility of the study.

Addressing the suggestions above would further strengthen the manuscript and enhance its value for researchers, policymakers, and practitioners.

**Do you want your identity to be public for this peer review?** For information about this choice, including consent withdrawal, please see our Privacy Policy

Reviewer #1: **Yes:** Junyan Liu

Reviewer #2: **Yes:** Songmei Lin

---

## [Author Response · Author response to Decision Letter 1]

17 Dec 2025

We have uploaded a file with our response to reviewers.

---

## [Decision Letter · Decision Letter 1]

12 Jan 2026

Dear Dr. Brussoni,

Thank you for submitting your manuscript to PLOS ONE. After careful consideration, we feel that it has merit but does not fully meet PLOS ONE’s publication criteria as it currently stands. Therefore, we invite you to submit a revised version of the manuscript that addresses the points raised during the review process.

We look forward to receiving your revised manuscript.

Kind regards,

Yih-Kuen Jan, PhD

Academic Editor

PLOS One

Journal Requirements:

**Additional Editor Comments:**

- References should not be cited in the conclusion section.

Reviewers' comments:

Reviewer's Responses to Questions

**Comments to the Author**

Reviewer #1: All comments have been addressed

Reviewer #2: All comments have been addressed

2. Is the manuscript technically sound, and do the data support the conclusions?

Reviewer #1: Yes

Reviewer #2: Yes

3. Has the statistical analysis been performed appropriately and rigorously?

Reviewer #1: Yes

Reviewer #2: Yes

4. Have the authors made all data underlying the findings in their manuscript fully available?

Reviewer #1: Yes

Reviewer #2: Yes

5. Is the manuscript presented in an intelligible fashion and written in standard English?

Reviewer #1: Yes

Reviewer #2: Yes

Reviewer #1: Thank you for submitting the revised manuscript. The revision is improved and clearer than the previous version. Overall, the study is technically sound and the conclusions are supported within the stated limits.

Reviewer #2: 1.Overall evaluation

The manuscript has been substantially improved following the authors’ careful and thorough revisions. The quality, clarity, and academic rigor of the paper have been significantly enhanced.

2.Remaining issues to be addressed

The manuscript still requires some minor revisions. First, the overall length of the paper should be reduced. In particular, the Discussion and Limitations sections are overly lengthy and would benefit from more concise and focused discussion that directly reflects the key findings of the study. Second, the Conclusions section is too verbose and should be further refined. The conclusions should clearly and succinctly summarize the main findings, practical implications, and directions for future research. In addition, references should not be cited in the Conclusions section.

**Do you want your identity to be public for this peer review?** For information about this choice, including consent withdrawal, please see our Privacy Policy

Reviewer #1: **Yes:** JUNYAN LIU

Reviewer #2: No

---

## [Author Response · Author response to Decision Letter 2]

18 Jan 2026

Thank you for the additional feedback. We have condensed the Discussion, Limitations and Conclusion sections. We have also removed references in the conclusion section.

---

## [Editor Report · Decision Letter 2]

21 Jan 2026

Examining outdoor play associations in Canadian early learning and child care centres: Cross-sectional insights from the Measuring Early Childhood Outside survey

PONE-D-25-43967R2

Dear Dr. Brussoni,

We’re pleased to inform you that your manuscript has been judged scientifically suitable for publication and will be formally accepted for publication once it meets all outstanding technical requirements.

Kind regards,

Yih-Kuen Jan, PhD

Academic Editor

PLOS One
---

## [Editor Report · Acceptance letter]

PONE-D-25-43967R2

PLOS One

Dear Dr. Brussoni,

I'm pleased to inform you that your manuscript has been deemed suitable for publication in PLOS One. Congratulations! Your manuscript is now being handed over to our production team.

Kind regards,

on behalf of

Dr. Yih-Kuen Jan

Academic Editor

PLOS One